# Current Issues on Research Conducted to Improve Women’s Health

**DOI:** 10.3390/healthcare9010092

**Published:** 2021-01-17

**Authors:** Charalampos Siristatidis, Vasilios Karageorgiou, Paraskevi Vogiatzi

**Affiliations:** 1Assisted Reproduction Unit, Second Department of Obstetrics and Gynecology, Medical School, National and Kapodistrian University of Athens, Aretaieion Hospital, 76 Vass Sofias, 11528 Athens, Greece; 22nd Department of Psychiatry, Medical School, National and Kapodistrian University of Athens, Attikon Hospital, 1 Rimini Street, 12642 Athens, Greece; vaskarageorg@hotmail.com; 3Andromed Health & Reproduction Diagnostic Lab, 3 Mesogion Str, 15126 Maroussi, Greece; evivogiatzi@gmail.com

**Keywords:** research quality, methodology, evidence-based medicine, systematic review

## Abstract

There are varied lessons to be learned regarding the current methodological approaches to women’s health research. In the present scheme of growing medical literature and inflation of novel results claiming significance, the sheer amount of information can render evidence-based practice confusing. The factors that classically determined the impact of discoveries appear to be losing ground: citation count and publication rates, hierarchy in author lists according to contribution, and a journal’s impact factor. Through a comprehensive literature search on the currently available data from theses, opinion, and original articles and reviews on this topic, we seek to present to clinicians a narrative synthesis of three crucial axes underlying the totality of the research production chain: (a) critical advances in research methodology, (b) the interplay of academy and industry in a trial conduct, and (c) review- and publication-associated developments. We also provide specific recommendations on the study design and conduct, reviewing the processes and dissemination of data and the conclusions and implementation of findings. Overall, clinicians and the public should be aware of the discourse behind the marketing of alleged breakthrough research. Still, multiple initiatives, such as patient review and strict, supervised literature synthesis, have become more widely accepted. The “bottom-up” approach of a wide dissemination of information to clinicians, together with practical incentives for stakeholders with competing interests to collaborate, promise to improve women’s healthcare.

## 1. Introduction

Women’s health has been at the center of interest and growing concern in the last few decades. As a measurable outcome, it has been studied at the level of mortality [1], serious morbidity [2], and nutritional status [3] and through proven, evidence-based interventions. The implementation of such interventions is essential to guide national and international policies and programs, targeting the achievement of universal coverage of health services. In this respect, conducting the best quality of research (research that provides firm and ethical evidence adhering to the principles of professionalism, transparency, and auditability) with the use of robust methods is mandatory. Towards this goal, the current reality is far from encouraging.

In accordance with scientific literature guidelines and research quality guidelines (e.g., the Preferred Reporting Items for Systematic Reviews and Meta-Analyses (PRISMA)), impact factors, and citation count are considered the norms in current research evaluation modalities. However, recent works in research methodology challenge this simplifying notion [4,5,6].

The pitfalls reported are associated with various—albeit specific—“cultural, ethical, operational, regulatory, and infrastructural factors” linked with a lack of adequately trained researchers and subject attrition bias [7]. As a result, the clinical research environment is more or less inseminated with various types of bias, leading to the discouragement of sponsors. The growing plethora of questionable quality trials and reviews is another issue to consider. “From 14 reports of trials published per day in 1980 [to] 75 trials and 11 systematic reviews of trials per day, a plateau in growth has not yet been reached”, stated a policy forum article reported in 2010 [8]; additionally, the authors noted that “the staple of medical literature synthesis remains the non-systematic narrative review”, further pointing out the need for freely available simple yet valid answers to most patients’ questions [8]. In the same context, current data concerning women’s health derived from protocols, full study reports, and participant-level datasets are rarely available to a wide audience. At the same time, selective reporting of methods and results plagues reports. With the reduced quality of information produced, a lot of money has been wasted; subsequently, the existence of all kinds of bias affect the research itself and jeopardize the validity of the findings and, consequently, the care of women [9].

Issues have been raised by exploring different approaches to evaluate the quality of scientific input to the community. The ultimate goal remains a robust and uniform literature evaluation system adapted to the evolving conduct of studies and to the application of modern tools to re-ensure robust methodology and reporting of data and results. Here, we perform a narrative overview on current issues in study quality assessment regarding clinical medicine. We electronically searched PubMed using the following keywords: “clinical trials”, “meta-analysis”, “IPD”, “sponsor”, “challenges”, “regulatory”, “women’s health”, “evidence-based medicine/trends”, “policy making”, “publishing”, “research methods and practices”, “consumers network”, “bias”, “industry-sponsored trials”, “biomedical bibliographic databases”, and “quality research”, trying to collect data irrespective of type of report and language. Based on this evidence, we propose a combination of interventions at various levels, underlining quality aspects that we consider significant, and other routes of judgments.

## 2. The “Standard” Factors

Citation count and publication rates in international databases, hierarchy in author lists according to contribution, and the impact factor of the journal are considered important factors in the quality of a study, especially for the “scientific reader” seeking quality information on a specific topic. This has been extensively studied by other workgroups [10,11,12,13] and represents a justified trend accounting for the prestige of a scientific journal and the publication itself, along with language and availability, and ultimately skewing scientific trends or potentially leaving some important contributions in obscurity. Even though previous works contemplate the importance of the aforementioned factors in the true quality of research studies and publications, all considerations are derived from a common denominator, that is, that the currently used quality standards either for the common user or for greater structures and institutions most probably do not reflect quality but rather popularity. In this context, the citation rate (including self-citation and “negative citing”) and an impact statement on the individual author (a concise summary of the impact of somebody’s career) have been proposed. In addition, other metrics, including altmetrics, bibliometrics, and H-index, combined with updated mathematical models, such as artificial neural networks, might be the tools of the future; these models constitute more accurate tools due to the special characteristics of these “learning through training” processes, resembling the capacity of the brain to learn and judge [14].

## 3. The Type of Research Question and Studies

Although multiple outcomes may be reported at once and variability in study designs fluctuates, a primary role belongs to the type of research question explored by a study or publication, which will inevitably determine the methodology to be followed. For example, in past years, there is a disproportionate output of Systematic Reviews (SRs) and meta-analyses from Asian countries produced on a massive scale [15,16] as a means of “publishing in order to publish” with questionable quality and methods. Their numbers are so high that, in some cases, it overtakes original trials. Of note, the use of such studies in the biomedical field was occasional until the 1990s [17]. Moreover, those from the Cochrane collaboration, the fundamental organization for good quality systematic reviews, are only a small fraction of this output [8]. 

With regard to Randomized Controlled Trials (RCTs), suggestions have been made in recent reports on better conduct [18]: trial protocols should be simple, reproducible, and well organized, with predefined and well-described study populations/participants and should have sound interventions, and representative comparisons and outcomes. Of note, these could be based on the conclusions of previously conducted SRs that often point out issues in quality and methodology of the original trials. Minimal deviations from protocols and a priori specification of useful core outcomes that translate directly to women’s wellbeing are the focus of the CROWN Initiative [19]. According to the authors, there has been a multi-targeted set of suggestions to “ensure that critical and important outcomes with good measurement properties are incorporated and reported, in advancing the usefulness of research, in informing readers, including guideline and policy developers, who are involved in decision-making, and in improving evidence-based practice”.

A priori description of the outcomes of interest can alleviate the known issues/biases associated with exploratory analyses. A change in outcome, especially in cases where the results do not support the rationale of the study, can mask the original intentions of the authors and can recontextualize the same results in a more positive manner [20]. Still, an exploratory analysis has a significant role in deducing potentially valuable conjectures for future studies. However, it is central for transparency that the authors explicitly state when this is the case, i.e., when an analysis is conducted post hoc. In order to ease the distinction of post hoc and a priori analyses by SR authors and readers, Dwan et al. (2014) proposed the publication of both protocols and pre-specified analyses [21].

We cannot anticipate that SRs can retrospectively solve the potential gaps and inconsistencies in the methodology and outcome reporting. For robust answers, research questions must be well defined from the start. However, more elaborate techniques of evidence synthesis can guide future research in more meaningful ways and are becoming more popular. Specifically, prospective and individual patient data meta-analyses (IPDMA) may need to become the norm in literature synthesis [22,23]. A major difficulty in IPDMA is the fact that securing sensitive patient data is a time-consuming task that demands the establishment of mutual trust. Even when representative evidence has been secured, data availability may still affect the pooled evidence. A recent study assessing IPDMA’s treating oncological topics suggested that studies for which they were available differed significantly from studies in which the authors did not share them [24]. Still, IPDMA is a trustworthy methodology that can assess the effect of patient-level covariates on treatment outcomes or diagnostic accuracy more thoroughly than the standard procedure of a meta-regression used in aggregate-data meta-analyses [25]. Given the current ethos of openness in clinical trials and common repositories becoming more widespread, IPDMA is likely to become the mainstay of critical synthesis of literature [26].

Finally, we have to include observational research in an effort to improve women’s health in the context of greater personalization of care and stratified medicine. Such studies have traditionally served as tools for understanding the nature of particular clinical conditions, for determining risk factors and mechanisms of actions, and for identifying potential intervention targets. Their disadvantages associated with methodological issues such as confounds and the fact that they are prone to limited internal validity could be restricted through guidelines such as the strengthening the reporting of observational studies in epidemiology (STROBE) statement [27].

## 4. The Inclusion of Young Authors

The encouragement of younger and/or less experienced scientists and ultimately their inclusion in the respective workgroups and in the list of contributors may provide an unexpected topic or question and a clearer view on established research schemes. The productivity of highly cited papers is related to the advanced age of their authors; adversely, better funding opportunities for younger researchers would give them unique chances to build strong productivity [28]. The advancement of knowledge taught during academic training as well as a higher probability of compliance with robust methods of reporting should encourage the inclusion of younger scientists. Tips and recommendations of young authors and early career scientists have been plenty, including collaborating with researchers within as well as outside their field and/or country, sending their research article to an appropriate journal, and adequately highlighting the novelty and impact of their research [29,30].

Towards this goal, the improvement of the scientific literacy of young scholars is the main step, and this burden falls on to the shoulders of the trainers. There are “uncomfortable truths” in training [31], but scientific research and the mode of thinking are processes continuously accumulated and must be taught by each director or responsible authority: they should improve the skills and capabilities of young scholars in scientific and technological literacy and in communication and productivity.

## 5. Quality in Reporting

Reporting quality must be ensured by avoiding bias, such as selective reporting, deliberate or not. Avoiding reporting insignificant data and outcomes could lead to severe distortion in the SR [32]. Thus, flaws in design, conduct, analysis, or reporting of RCTs can produce bias in the estimates of a treatment effect. 

For example, in a large meta-epidemiological study of 1973 RCTs, a lack of blinding was associated with an average 22% exaggeration of treatment effects among trials that reported subjectively assessed outcomes [33]. This deviation is enough to adversely affect the interpretation of the results and further negatively influences regulatory settings and clinical practice.

Another example involves the evidence base on recent cancer drug approvals. Between 2014 and 2016, a quarter of the relevant studies were not RCTs; of the RCTs, the majority of them did not measure overall survival or quality of life outcomes as primary endpoints, and half of them were judged to be at high risk of bias; the authors’ judgments changed for a fifth of them when they relied on information reported in regulatory documents and scientific publications separately [34]. 

## 6. Strict Implementation of Rules in the Peer Review Processes

These processes first appeared in 1655 in a collection of scientific essays by Denis de Sallo in the *Journal des Scavans*, and almost 100 years later (1731), their implementation became a standard of practice by almost all biomedical journals [35]. Maintaining the quality and scientific integrity of publications; evaluating for competence, significance, and originality; and ensuring internal and external validity of submissions are crucial points. Similarly, the appropriate selection and training of reviewers to provide quality and specialized reviews without bias is an essential part of the process [36].

## 7. Sponsorship

Ethical and other issues surrounding sponsorships, to ensure credibility of a study, have been addressed in the past. One of the main sources of funding remains the industry [37]. Indeed, sponsored clinical research has always been questioned, influenced by reports of selective or biased disclosure of research results, ghostwriting and guest authorship, and inaccurate or incomplete reporting of potential conflicts of interest [38]. Although these may be a scarce incidence nowadays, active monitoring in funded studies should be implemented throughout in order to eliminate this possibility or any other conflicts of interest. An alarming analysis of 319 trials indicated that only a small minority (three out of 182 funded trials) were funded by multiple sponsors with competing interests. The presence of industry funding also almost tripled (OR = 2.8, 95% CI: 1.6, 4.7) the possibility of a study having reported favorable findings [39]. Furthermore, registered study protocols that announced funding were less likely to be published after their completion (non-publication rate: 32% vs. 18% [40].

## 8. Change in the Notion of Publishing

The change in notion and perception of the impact of outcomes is perhaps the most important part of the improvement in research conduct and implementation. This can be achieved through differentiation and modern adaptation of our scientific culture fighting inner and external incentives. Every scientific input should target a wider human benefit. A change in notion and incentives in publishing is crucial, from the level of the investigator aiming to publish/individual behaviors up to the social forces that provide affordances and incentives for those behaviors [41,42,43].

In a specific area of research, a clinical evaluation should precede publication in order to ensure relevance. A dramatic example is the scientific literature demonstrating an overload of various biomarkers for various diseases in which only a few of them have been confirmed by subsequent research and few have entered routine clinical practice [44]. In addition, biomarkers should also be judged on the grounds of cost-effectiveness and incremental net benefit [45]. Multiple indices may have comparable diagnostic accuracy, but their cost, an unavoidable concern in public health, may differ significantly.

Therefore, the selection of information to be published should be conducted on safer grounds and should be adequately supported by the authors, based on our knowledge on the scheme to date, and importantly, a summary of previous attempts should note the effective interventions and provide a concluding remark for the scientist through a good quality review.

## 9. Patient’s Contribution to Evaluation and Sex/Gender Analyses

In the era of evidence-based medicine, feedback from the recipients of healthcare development is gaining more importance and platforms for opinion exchange between patients and investigators have been established. This has already been implemented by the Cochrane Collaboration, where patient review is an integral part of the SR publication process and plain language summaries target a nontechnical audience. This process could be adopted as standard practice if accordingly modified. If the patient review, for example, is to be widely implemented in other journals, it would constitute a potentially radical paradigm shift that aims to solidify the review process. Of course, technical difficulties, such as acknowledgement and incentives for patients participating in review processes, are fields where further developments will enhance this policy [46].

It has been noted that the women population represents an “unequal majority” in health and health care. It is also well established that women’s health needs are dissimilar from those of men, resulting from the fact that both the woman’s body and brain functions differ critically from a man’s and that she reacts differently to even the same stimuli, such as medications or environmental events. It is indicative that, even though a large proportion of study protocols included women, only 3% of them planned an analytical approach for quantifying sex differences [47]; similarly, a recent report on therapies for atrial fibrillation concluded that the sex-specific reporting in trials comparing them was extremely low [48]. As a result, women have not received an ideal “personalized” health care, in many cases, so far. Thus, a specific design for studies on women’s health should be required.

There are several examples in the history of women’s health research where the contribution of the consumer women’s health movement in promoting research in women’s interests was critical. One of them concerned the collaborations between consumer groups and researchers in obtaining funding in the U.S. and France for a follow-up on a cohort of diethylstilboestrol-exposed people when the drug was discovered to be a transplacental carcinogen in pregnancy in 1971.

Another important issue is the nonavailability of sex/gender data from primary studies and consequently from SRs, which are the main tools to provide the necessary evidence for the formation of relevant policies [49]: the authors stated that even “Cochrane and the Campbell Collaboration have no specific policy on the reporting of sex/gender in systematic reviews, although Cochrane has endorsed the SAGER guidelines developed by the European Association of Science Editors” [50]. In their review, they found that the Methods sections of these collaborations included the most reports on sex/gender in both Campbell (50.8%) and Cochrane (83.1%) reviews, but the majority of these were descriptive considerations of sex/gender. They also reported that 62% of Campbell and 86% of Cochrane reviews did not report sex/gender in the abstract but included sex/gender considerations in a later section. A previous study on the subject reported that almost half of SRs described the sex/gender of the included populations but only 4% assessed sex/gender differences or included sex/gender when making judgments on the implications of the evidence [51]. 

## 10. An Improvement in the Dissemination of Studies

Despite advances in the dissemination of study information, half of health-related studies remain unpublished [52]. Problems in the publishing scheme in the selection of studies that appear to have a higher impact or that come from a respectable institution can lead to biased publishing. At the extreme, unsafe, ineffective, or even harmful interventions may enter clinical practice, as was the case with hormone replacement therapy [53]. In some instances, even a shift in healthcare resource allocation is reported [9]. It is standard practice in critical readings of literature to evaluate publication bias. This method attempts to address, with controversial success, precisely the unfortunate keenness of editors to promote positive results that imply novelty. A classic example of this inflation of positive and supposedly important results is the 2012 study by Fanelli [54], in which studies classified as related to clinical medicine showed a gradual increase in reporting positive findings. The author criticized the efficacy of measures taken to attenuate publication bias, e.g., protocol registration.

On the other hand, a respectable amount of research is published in other languages and not indexed in U.S. National Library of Medicine [55], while their quality remains controversial [56]; the authors of the above studies stated that peer review processes need to be improved through guidelines aiming to identify the authenticity of the studies.

The bulk of peer reviews remain a voluntary occupation, with the main motivation being recognition by peers. In addition, statistical review, a time-consuming process, is not performed in all published research. This process can be accelerated by practices that promote data and code sharing. It is also suggested that, even when papers are retracted, this could have been avoided with the simple measure of an active data sharing policy [57].

## 11. Role of the Stakeholders and Foundations

For the stakeholders and collaborative systems, a more energetic role is required in ensuring the conduct of multicenter massive-trials with increased clinical relevance. The main problem in the conduct of research is the lowered clinical value of the results from small sample sizes, even in RCTs. Mathematical models have been developed to predict sample sizes corresponding to the clinical value of the outcomes, while patient data from databases could easily increase the sample size of trials at much lower costs. Such paradigms could include the Health Care Systems Research Collaboratory and the Patient-Centered Outcomes Research Network (PCORnet) [58]. Also, new levels of patient engagement can raise the possibility of improving clinical outcomes on health. Involving multiple stakeholders (with potentially conflicting interests) in shared conversations on research has been proposed [59].

New foundations should be placed in research by focusing on the improvement of quality, such as NIH and PCORI [60]. The Cochrane Collaboration represents one of the very few large-scale initiatives in this context; importantly, both conduct high quality reviews, and participant education at all levels are based mostly from volunteers who care about science and high-quality evidence.

## 12. Cooperation of All Forces: The Role of Industry/Funding

The central point of problem is funding. USA-affiliated industry-funded trials and related activities represent more than 5% of US healthcare expenditure, with approximately $70 billion in commercial and $40 billion in governmental and non-profit funding annually. The NIH invests $41.7 billion annually in medical research: 80 percent is awarded for extramural research, through 50,000 competitive grants to more than 300,000 researchers at more than 2500 universities, medical schools, and other research institutions [61]. Concerns have been raised that this approach appears inefficient for how biomedical research is chosen, designed, regulated, financed, managed, disseminated, and reported [62,63,64].

The scheme, however, has been shifting in favor of Asian countries. Factors, such as ease of recruitment, population, and various epidemiological factors (e.g., increased incidence of infectious disease) have contributed positively to an inflation of local clinical trials [7]. Severe accusations regarding clinical data management have been raised, although the magnitude of the problem cannot be safely evaluated [65]. This unavoidably hinders the validity and future usefulness of these results despite initial enthusiasm from editors and the industry.

Economic forces are important, and ultimately, the industry seeks to maximize profit by providing new products and services to the medical market [66]. In industry-funded clinical research, intentional and unintentional commercial motives can control the study design and comparators. Governmental involvement [66] has an important role in distributing research funds in areas important for the protection and restoration of human health, even when the prospects for commercial profit are poor or nonexistent. The recruitment of specialized and qualified professionals should set higher standards of rigor when they are involved in commercial or unavoidably conflicted relationships and to disseminate the resources evenly, especially when nowadays these are scarce.

Funders and academic institutions are responsible for the moral status, as research usually initiates from there and determines any kind of shift in the process. Academics might be judged on the methodological rigor and full dissemination of their research, the quality of their reports, and the reproducibility of their findings. Previous reports suggest ways to increase the relevance and to optimize resource allocation in biomedical research, indicating how resource allocation should be conducted, along with revisions in the appropriateness of research design, methods, and analysis, with efficient research regulation and management fully accessible information, promoting unbiased and usable reports. Additionally, motivation must be given to authors to share their data [67], as has been performed in the field of genetics [68]. Of note, synthesis of evidence on the meta-epidemiological level cannot always confidently provide answers to practical clinical questions [69]. 

Compromised ethics should be traced and removed from independent research and academia, while journals should on no occasion put profit and publicity above quality. The solution to this lies on the progressive refinement of methods and improvement of the objective and controlled processes.

## 13. Training

Essential training and interprofessional learning of clinicians and other hands-on scientists in the medical field are an absolute must. There is a growing need to improve their scientific insight and judgment. Reviewers should learn how to apply an unbiased critical thinking and evaluation of the methods explored, of the study questions, and of the resulting impact towards good clinical practice and human welfare. This not only applies to organizational refinement by the Academic Institutes and Publishing Organizations but also to the scientists themselves to obtain the drive to train, along with methodologists and statisticians, so that specialization and knowledge is shared and every contributor works soundly towards a common cause. 

## 14. Conclusions

Research is a solid foundation for the progression of sciences, and the key importance in maintaining the evolution of knowledge is “contributing and sharing”, but this has to be performed adequately. Although there are several criteria and controlled circumstances under which new data and overviews of data are published, research and publishing methods require continuous readjustments and modifications to ensure quality. An overview of the published literature on women’s health and its relevant subtopics is an excellent paradigm on a crucial field of the different types of research and publications that one may encounter but also an example of the vast variability in information available, not only in terms of results but also in terms of design, analysis, quality of information, and implementation of results. In clinical practice, it is imperative to assess information collectively a researcher, medical expert, funder, reviewer, and patient, and this should encourage the improvement of evidence-based patient management.

This review aimed to present the major nodal points of quality and to propose a combination of interventions at various levels, along with other routes of judgement. We also sought to address potential flaws and pitfalls in research conduct and to provide recommendations upon improvement of study designs/methods and scientific reporting to promote publication quality and stricter criteria for release with support from the appropriate structures. A summary of recommendations towards evidence implementation as presented in Table 1 could comprise valuable guidance to both the health experts and the health service recipients to which these standards are quality criteria. A meticulous study design that promotes the transparency of methods and potential conflicts allows a clear distinction of the pathologies and targeted groups and that provides substantial scientific background should be pursued by both researchers and readers. Robust implementation of the pre-stated methods and approaches of analysis, with active participation of collective fronts tied to the subject, should allow quality output to be published and should add value to the findings. Patient-first and common welfare should be considered throughout in conjunction with supporting and providing evidence on robust outcomes for the improvement of healthcare, that may be facilitated by healthy and network collaborations. 

How these recommendations should be accounted for, evaluated, and implemented relies on the individual discretion of the reader, the scientist, the author, or any entity affiliated with a publishing organization and should be customized to be applied individually for each specialized academic and scientific field but also tailored across continents and countries. The latter is derived from the realization that research conduct, funding, and even the monitoring authorities of clinical studies rely on nonuniform procedures among countries and unions and conforms to different legal frameworks across countries. Nevertheless, a core of actions, precautions, and a quality exemplar of golden standards should be constructed and widely applied to meet the standards that describe a representative scientific contribution, for example, uniform, widely accepted, and practiced standards through policies, guidelines, and rules on a national and/or international level created either by in-country legislation or by scientific entities; allocation of the resources for their implementation; and mechanisms of control for their application and adherence by all.

In conclusion, multiple steps throughout the long and costly process of trial conduct are prone to bias. Notably, increasing international competition favors faster and cheaper patient recruitment, conduct, and analysis and, in turn, produces questionable research. Literature synthesis through SRs and/or meta-analysis has a primarily retrospective role that guides future research and sheds light on arguable topics but cannot erase the wrongdoings of primary studies, which are often concealed. The “bottom-up” approach of a wide dissemination of information to clinicians, together with practical incentives for stakeholders with competing interests to collaborate, promise to improve women’s healthcare.

## Figures and Tables

**Table 1 healthcare-09-00092-t001:** Summary of the recommendations for the steps towards evidence implementation.

Domain	Recommendations
**Study Designs**	Declaration of competing interests
	Prospective and individual patient data meta-analyses
	Incorporation of evaluated observational research
	Inclusion of women and explicit analysis of gender differences
**Study Conduction**	Pre-specification of outcomes
	Abidance by protocols
	Transparency in reporting
	Inclusion of young trained authors
**Review**	Training and motivations for peer reviewers
	Active participation of patients
	Active monitoring in funded studies
	Stricter criteria to promote publication quality
**Dissemination**	Publication of both negative and positive results
	Clinical evaluation should precede publication
	Every scientific input should target a wider human benefit
	Groups to check for unpublished data
	Multiple stakeholders cooperation
**Implementation of Findings**	Training of clinicians
	Critical attitude towards

## Data Availability

Not applicable.

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
