# Peer review of "Current Issues on Research Conducted to Improve Women’s Health"

_healthcare, 2021, doi:10.3390/healthcare9010092_

Round 1

Reviewer 1 Report

In this concept paper, authors provided some views on how to improve the research quality, which are helpful for readers. They also presented some critical issues on flaws and pitfalls related to academic studies, which are important topics in academic community. However, if possible, the following comments should be addressed further:

  1. About impact factors of academic journals, authors pointed out its drawback but they did not discuss further how scientists, readers or public use it rationally. I mean author just point out this issue, but did not present potential solutions.
  2. About the type of study, author focused on Meta-analysis and RCT, but they seemed to ignore the observational studies, which are one of most important type of study. Although these observational studies cannot assess the cause-effect directly, they serve as important tool to look for risk factors or etiological clues of diseases. Actually, there are lots of scientific issues concerned when conducting or reporting observational studies. At present, lots of meta-analysis are based on observational studies. How to find or control key issue in conducting observational studies is still problem in practice even though some guidelines as STROBE are available.
  3. The encouragement of younger and/ or less experienced scientists and ultimately their inclusion in the respective workgroups is important. However, it may be more important to improve the scientific literacy of young scholars. Their minds are active, but they need more scientific guidance. Scientific research and mode of thinking is a process of continuous accumulation.
  4. In this paper, authors use a lot of space to discuss the issues related to the research rather than the research itself, which might not be consistent with title. A better title might be needed.
  5. I disagree with authors about sex/gender analyses. Some SR publications cannot present sex/gender difference largely because design of original studies did not consider sex/gender so that sample size estimation is just based on overall population not by sex/gender. Therefore, specific design of study on woman or child health should be required.

Author Response

Reviewer no1

In this concept paper, authors provided some views on how to improve the research quality, which are helpful for readers. They also presented some critical issues on flaws and pitfalls related to academic studies, which are important topics in academic community. However, if possible, the following comments should be addressed further:

  1. About impact factors of academic journals, authors pointed out its drawback but they did not discuss further how scientists, readers or public use it rationally. I mean author just point out this issue, but did not present potential solutions.

A: We thank the reviewer for the comment. We have added two sentences in the end of the no2 section, to address this. It now reads:”In this context, the citation rate (including self-citation and “negative citing”) and an impact statement of an individual author (a concise summary of the impact of somebody’s career) have been proposed. In addition, other metrics, including altmetrics, bibliometrics and H-index combined with updated mathematical models, such as artificial neural networks might be the tools of the future; these models constitute more accurate tools due to their special characteristics through ‘learning through training’ processes, resembling the capacity of the brain to learn and judge (Siristatidis et al. 2020).”.

  1. About the type of study, author focused on Meta-analysis and RCT, but they seemed to ignore the observational studies, which are one of most important type of study. Although these observational studies cannot assess the cause-effect directly, they serve as important tool to look for risk factors or etiological clues of diseases. Actually, there are lots of scientific issues concerned when conducting or reporting observational studies. At present, lots of meta-analysis are based on observational studies. How to find or control key issue in conducting observational studies is still problem in practice even though some guidelines as STROBE are available.

A: We have added a paragraph in the end of the no3 section, to address this. It now reads:”Finally, we have to include observational research in an effort to improve women’s health, in the context of greater personalization of care and stratified medicine. Such studies have been traditionally served as tools for understanding the nature of particular clinical conditions, determining risk factors, mechanisms of actions and for identifying potential intervention targets Their disadvantages associated with methodological issues such as confounding and the fact that they are prone to limited internal validity could be restricted through guidelines, such STROBE (Barnish et al 2017).”. We have also incorporated this in the Figure.

  1. The encouragement of younger and/ or less experienced scientists and ultimately their inclusion in the respective workgroups is important. However, it may be more important to improve the scientific literacy of young scholars. Their minds are active, but they need more scientific guidance. Scientific research and mode of thinking is a process of continuous accumulation.

A: We have added a paragraph in the end of the no3 section, to address this. It now reads:”Towards this goal, the improvement of the scientific literacy of young scholars is the main step; and this burden falls on to the shoulders of the trainers. There are “uncomfortable truths” on training (Siristatidis et al 2011). Scientific research and mode of thinking is a process of continuous accumulation and the job of each director or responsible authority: they should improve skills and capabilities on scientific and technological literacy, along with communication and productivity issues.

  1. In this paper, authors use a lot of space to discuss the issues related to the research rather than the research itself, which might not be consistent with title. A better title might be needed.

A: The title has been changed to “Current issues on research conduction towards improved women’s health”.

  1. I disagree with authors about sex/gender analyses. Some SR publications cannot present sex/gender difference largely because design of original studies did not consider sex/gender so that sample size estimation is just based on overall population not by sex/gender. Therefore, specific design of study on woman or child health should be required.

A: We have added a paragraph in the middle of the no9 section, to address this. It now reads:”It has been noted that women population represents an “unequal majority” in health and health care. It is also well established that women’s health needs are dissimilar from those of men, resulting from the fact that both thewoman’s body and brain functions differ critically from a man’s, and that she reacts differently to even the same stimuli, such as medications or environmental events. Thus, specific design of study on woman health should be required”.

Reviewer 2 Report

This is a useful analysis which takes a unique look at improved women’s health through higher quality research conduction and implementation of robust methods. The core findings of the analysis is interesting, however there are some concerns needed be addressed:

ABSTRACT

" Factors that classically determined the impact of discoveries appear to lose ground”. This is somewhat vague. A couple of factors should be mentioned. The result or study findings seems to be missing in this section. I will suggest the study findings are mentioned.

INTRODUCTION

" In this respect, the conduction of the best quality of research is mandatory, linked with the use of robust methods" – I suggest you explain to your audience what you consider as best quality of research.

“Bearing also in mind that women have received a less-than-ideal health care, this becomes more pertinent” – This is a little misleading. This statement is not clear. I suggest you expatiate on it for better clarity.

“It is indicative that, even though a large proportion of study protocols included women, only 3% of them planned an analytical approach for quantifying sex differences (Holdcroft N 2007)” This study was carried out by the author about 14 years ago are there no current literature which provides a more recent scenario?

CONCLUSIONS

The last point (14): These paragraph(s) should focus more on synthesizing the key findings of the study, highlighting the most important findings. I will suggest this paragraph should be more elaborate than what it currently is. After the study findings synopsis readers will want to see the many potential sources of bias in the study and if these biases could have distorted results in any way.

Author Response

Reviewer no2

This is a useful analysis which takes a unique look at improved women’s health through higher quality research conduction and implementation of robust methods. The core findings of the analysis is interesting, however there are some concerns needed be addressed:

ABSTRACT

" Factors that classically determined the impact of discoveries appear to lose ground”. This is somewhat vague. A couple of factors should be mentioned. The result or study findings seems to be missing in this section. I will suggest the study findings are mentioned.

A: We thank the reviewer for the useful comments. We agree. We have now added 2 sentences in the abstract section to address these issues.

“Factors that classically determined the impact of discoveries, such as citation count and publication rates, hierarchy in the authors list according to contribution and journal’s impact factor, appear to lose ground.

We also provide specific recommendations on study design and conduction, reviewing processes and dissemination of data and conclusions and implementation of findings.”.

INTRODUCTION

" In this respect, the conduction of the best quality of research is mandatory, linked with the use of robust methods" – I suggest you explain to your audience what you consider as best quality of research.

A: We thank the reviewer for the useful comments. We agree. It now reads:”best quality of research (research that provides firm and, ethicalevidence, adhering to principles of professionalism, transparency, and auditability)is…”.

“Bearing also in mind that women have received a less-than-ideal health care, this becomes more pertinent” – This is a little misleading. This statement is not clear. I suggest you expatiate on it for better clarity.

A: We thank the reviewer for the useful comments. We agree. It now reads:”It has been noted that women population represents an “unequal majority” in health and health care. It is also well established that women’s health needs are dissimilar from those of men, resulting from the fact that both thewoman’s body and brain functions differ critically from a man’s, and that she reacts differently to even the same stimuli, such as medications or environmental events. It is indicative that, even though a large proportion of study protocols included women, only 3% of them planned an analytical approach for quantifying sex differences (Holdcroft N 2007).As a result, women have not received an ideal “personalized” health care, in many cases, so far.Thus, specific design of study on women’s health should be required.

“It is indicative that, even though a large proportion of study protocols included women, only 3% of them planned an analytical approach for quantifying sex differences (Holdcroft N 2007)” This study was carried out by the author about 14 years ago are there no current literature which provides a more recent scenario?

A: We thank the reviewer for the useful comment. We have added a more recent reference; it now reads:”…; similarly, a recent report on therapies for atrial fibrillation, concluded that the sex-specific reporting in trials comparing them was extremely low(du Fay de Lavallaz et al 2019).”.

CONCLUSIONS

The last point (14): These paragraph(s) should focus more on synthesizing the key findings of the study, highlighting the most important findings. I will suggest this paragraph should be more elaborate than what it currently is. After the study findings synopsis readers will want to see the many potential sources of bias in the study and if these biases could have distorted results in any way.

A: We thank the reviewer for the comment. We have thoroughly amended this section to address the issues raised. Please see revised text.

Reviewer 3 Report

The paper is well written, clear and mostly grammatically correct.

An interesting concept paper on women's health research.

Of interest to women's health researchers

I think the authors should refer to 'study conduct' not study conduction

Conduction is  an inappropriate word in this context throughout the paper.

Author Response

The paper is well written, clear and mostly grammatically correct.

An interesting concept paper on women's health research.

Of interest to women's health researchers

I think the authors should refer to 'study conduct' not study conduction

Conduction is  an inappropriate word in this context throughout the paper.

A: We thank the reviewer for the useful comment. We have changed the word throughout the text.

Reviewer 4 Report

The paper's authors have set out to explore the very significant issue of how to assess and improve the soundness and reliability of methodological approaches in medical research.

There is no denying that the paper does lay out a broad array of complex, multifaceted issues that need to be accounted for when evaluating research quality; it is in that respect worthy of being considered for publication, in light of the painstaking analytical endeavor the authors have embarked upon.

Still, it needs to be proof-read thoroughly in order to improve clarity and overall readability; questionable vocabulary choices are often found throughout. Those aspects need to be improved.

For instance:

Abstract, line 4: "...evidence-base practice..."

Abstract, line 4-5: "Factors that classically determined the impact of discoveries appear to lose ground." To be losing ground would be more in line with the intended meaning. 

Page 1: "Towards this goal, the today’s reality is rather different."

Page 5: "Another important issue is the absence of availability..." (better word it as "non-availability" or "unavailability", or dearth/shortage)

Page 6: "...authors stated that peer review processes need to be improved, through guidelines, aiming to identify the authenticity of the studies." What authors are you referring to? The ones cited above? Please add reference or clarify.

Page 6: "USA-affiliated industry funded trials and related
activities represent more than 5% of US health-care expenditure, with approximately $70 billion in commercial and $40 billion in governmental and non-profit funding annually; concerns have been raised that this approach appears inefficient as to how biomedical research is chosen, designed,
regulated, managed, disseminated and reported (Dorsey et al. 2010)." Please look for more recent data and add references; the analysis you referenced is quite old (2003-2008). To that end, you may want to consider: NIH funding and the pursuit of edge science, by Mikko Packalen and Jay Bhattacharya, or US biomedical and medical research under the Trump administration or Noble P, Ten Eyck P, Roskoski R Jr, Jackson JB. NIH funding trends to US medical schools from 2009 to 2018. PLoS One. 2020;15(6):e0233367. Published 2020 Jun 1. doi:10.1371/journal.pone.0233367

Page 7: Training of clinicians and other hands-on scientists to the medical field to see behind the scenes is an absolute must. ("...see behind the scenes" sounds awkward and quite unclear. Please rephrase.)

Page 8: "...even the monitoring authorities of clinical studies rely in non-uniform procedures". Rely on more like it.

Page 8: "Nevertheless, a core of actions, precautions and a quality exemplar of
golden standards should be constructed and widely applied to meet the standards that describe a representative scientific contribution." Please lay out in greater detail how the implementation of more uniform standards should be devised, overseen and guaranteed.  

Figure 1: Abidance with should be replaced with Abidance by.

Please consider finding more recent sources: the references you picked are quite old overall.

Author Response

The paper's authors have set out to explore the very significant issue of how to assess and improve the soundness and reliability of methodological approaches in medical research.

There is no denying that the paper does lay out a broad array of complex, multifaceted issues that need to be accounted for when evaluating research quality; it is in that respect worthy of being considered for publication, in light of the painstaking analytical endeavor the authors have embarked upon.

Still, it needs to be proof-read thoroughly in order to improve clarity and overall readability; questionable vocabulary choices are often found throughout. Those aspects need to be improved.

For instance:

Abstract, line 4: "...evidence-base practice..."

Abstract, line 4-5: "Factors that classically determined the impact of discoveries appear to lose ground." To be losing ground would be more in line with the intended meaning. 

Page 1: "Towards this goal, the today’s reality is rather different."

Page 5: "Another important issue is the absence of availability..." (better word it as "non-availability" or "unavailability", or dearth/shortage)

Page 6: "...authors stated that peer review processes need to be improved, through guidelines, aiming to identify the authenticity of the studies." What authors are you referring to? The ones cited above? Please add reference or clarify.

Page 6: "USA-affiliated industry funded trials and related
activities represent more than 5% of US health-care expenditure, with approximately $70 billion in commercial and $40 billion in governmental and non-profit funding annually; concerns have been raised that this approach appears inefficient as to how biomedical research is chosen, designed,
regulated, managed, disseminated and reported (Dorsey et al. 2010)." Please look for more recent data and add references; the analysis you referenced is quite old (2003-2008). To that end, you may want to consider: NIH funding and the pursuit of edge science, by Mikko Packalen and Jay Bhattacharya, or US biomedical and medical research under the Trump administration or Noble P, Ten Eyck P, Roskoski R Jr, Jackson JB. NIH funding trends to US medical schools from 2009 to 2018. PLoS One. 2020;15(6):e0233367. Published 2020 Jun 1. doi:10.1371/journal.pone.0233367

Page 7: Training of clinicians and other hands-on scientists to the medical field to see behind the scenes is an absolute must. ("...see behind the scenes" sounds awkward and quite unclear. Please rephrase.)

Page 8: "...even the monitoring authorities of clinical studies rely in non-uniform procedures". Rely on more like it.

Page 8: "Nevertheless, a core of actions, precautions and a quality exemplar of
golden standards should be constructed and widely applied to meet the standards that describe a representative scientific contribution." Please lay out in greater detail how the implementation of more uniform standards should be devised, overseen and guaranteed.  

Figure 1: Abidance with should be replaced with Abidance by.

Please consider finding more recent sources: the references you picked are quite old overall.

A: We thank the reviewer for the very detailed comments. We did our best to conform to all suggestions. Please find all changes in the revised (trough track changes) text.

Round 2

Reviewer 1 Report

Authors have addressed most of my concerns, and the manuscript has been improved much for publication.